# Modulation of initial leftward bias in visual search by parietal tDCS

**Laurie Geers**[1,2], **Valérie Dormal**[1,2], **Mario Bonato**[3], **Yves Vandermeeren**[2,4],
**Nicolas Masson**[1,2], **Michael Andres**[1,2]*

**1** Psychological Science Research Institute, UCLouvain, Louvain-la-Neuve, Belgium, **2** UCLouvain, Institute of Neuroscience (IoNS), NEUR Division, UCLouvain, Louvain-la-Neuve, Belgium, **3** Department of General Psychology, University of Padova, Padua, Italy, **4** CHU UCL Namur–Godinne Neurology Department, Stroke Unit & Neuromodulation Unit, UCLouvain, Louvain-la-Neuve, Belgium

* michael.andres@uclouvain.be

**Data Availability Statement:** The data that support the findings of this study are available in the Open Science Framework. Link for peer review: https://osf.io/4dbs5/.

## Abstract

Transcranial direct current stimulation (tDCS) has the potential to modulate spatial attention by enhancing the activity in one hemisphere relative to the other. This study aims to inform neurorehabilitation strategies for spatial attention disorders by investigating the impact of tDCS on the performance of healthy participants. Unlike prior research that focused on visual detection, we extended the investigation to visual search and visual imagery using computerized neuropsychological tests. Forty-eight participants had to actively search for targets in space (visual search) and notice differences between two mental images (visual imagery). Anodal stimulation was administered over the left parietal cortex for half of the participants and over the right parietal cortex for the other half. The results showed that tDCS modulated spatial attention in visual search but not in visual imagery. In the sham condition, visual search was characterized by a leftward bias in the selection of the first target and a left asymmetry in the overall spatial distribution of cancelled targets. Parietal tDCS modulated the initial leftward bias, enhancing it (more lateral) during right anodal stimulation and reducing it (more central) during left anodal stimulation. However, this effect was not reflected in the spatial distribution of the cancelled targets. The overall visual search performance marginally improved during right anodal stimulation, as evidenced by a greater percentage of cancelled targets compared to sham. Finally, the results revealed no left-right asymmetries in the visual imagery task, either after sham or anodal stimulation. The specific effect of parietal tDCS on the initiation of visual search offers a new perspective for targeted neurorehabilitation strategies and provides further insight into the different sensitivity of visual search measures classically used in brain-lesioned patients.

## 1. Introduction

Transcranial direct current stimulation (tDCS) allows modulation of human brain functions by causing the neural tissue to depolarize or hyperpolarize [1]. As a consequence of polarization, the spontaneous firing rate is increased under the anode and decreased under the

**Funding:** This research was supported by the following funding sources: the Fédération Wallonie Bruxelles (FWB), under grant ARC21/26-112, awarded to L.G.; the Fonds De La Recherche Scientifique – FNRS, under grant T.0245.16, awarded to M.A.; the Fonds Spécial de Recherche (FSR) at UCLouvain, under grant T.0047.18, awarded to M.A.; the Dipartimenti di Eccellenza DMm, under grant 11/05/2017 n. 262, awarded to M.B.; the Fonds De La Recherche Scientifique – FNRS, under grant 1.R.506.16, awarded to Y.V.; and the Fonds de la Recherche Scientifique Médicale (FRSM), under grant 3.4.525.08.F, awarded to Y.V.

**Competing interests:** The authors have declared that no competing interests exist.

cathode, presenting an opportunity to modulate abnormal activity and restore impaired brain functions with a portable, non-invasive device [2, 3]. While many studies focused on motor functions, our study examined visuospatial abilities whose deficit has been related to an imbalance in the activity of the two hemispheres. Unilateral spatial neglect occurs frequently after stroke, *e.g.*, in about 23–80% of patients [4–6], and has a negative impact on motor rehabilitation and functional autonomy [7, 8]. Neglect patients typically fail to respond to stimuli located on the side opposite to the brain lesion [9]. In the healthy brain, each hemisphere is in charge of appraising the contralateral side of space and reciprocal inhibitory inter-hemispheric connections allow for attention orientation [10, 11]. Assuming that unilateral spatial neglect is due to an imbalance of inter-hemispheric interactions caused by a release of inhibition from the damaged hemisphere to the undamaged hemisphere [12], tDCS should compensate neglect by enhancing neural activity in the damaged hemisphere or reducing neural activity in the undamaged hemisphere. In healthy individuals, anodal stimulation over the left or right parietal cortex improved visual detection in the contralateral hemifield, whereas cathodal stimulation impaired detection of the contralateral stimuli [13, 14]. Similarly, left anodal stimulation induced a rightward bias in greyscale judgements [15], whereas right cathodal stimulation induced a rightward bias in symmetry judgements of prebisected lines [16], or slowed down performance [17, 18]. It is worth noting that some studies also found an effect of tDCS on attention orientation that was independent of the hemifield [19, 20]. In stroke patients, studies evidenced a reduction of the ipsilesional spatial bias typically observed in line bisection after anodal stimulation of the damaged hemisphere and/or cathodal stimulation of the undamaged hemisphere [14, 21, 22].

However, previous studies focused on spatial biases in visual detection and line bisection, while the diagnosis and conceptualization of neglect often involves quantifying several other spatial abilities [23, 24]. We chose to examine the effects of parietal tDCS on two core spatial abilities, namely visual search and visual imagery. The choice of these two abilities is empirically motivated by evidence collected with neglect patients. First, searching an object among distractors, as typically assessed by "cancellation" tests, predicts clinical signs of neglect better than line bisection tests [25–27]. Cancellation tests are therefore widely used and considered as a clinical "gold standard" for detecting spatial asymmetries. Second, neglect patients may also experience difficulties in visualizing and describing the contralesional features of internally generated images of familiar places (*e.g.*, a house room, a well-known square or a country map [28–30]); and objects (*e.g.*, a clock face representing a given time [31, 32]). Clinical data suggest that representational neglect is as frequent as perceptual neglect [31] and some theoretical accounts build on the idea that the representational impairment plays a primary role in visual neglect [28, 33–35]. A third issue concerns the implementation of spatial attention tasks that could directly benefit to the assessment of patients. In clinical practice, performance in cancellation tests is generally summarized into a single score used as a binary classifier to identify the presence or absence of neglect (*e.g.*, number of left *vs.* right targets cancelled), while imagery neglect assessment is highly dependent on introspective verbal reports (*e.g.*, describe a well-known place). We used computerized tasks allowing a refined measurement of the distribution of spatial attention and a strict control of the visual display (*e.g.*, stimulus duration or speed). It is now recognized that paper-and-pencil tests, such as line bisection, lack sensitivity while attention-demanding computerized tasks may reveal neglect several years after stroke even when patients succeed in standard tests [36–39]. Understanding how tDCS interacts with spatial behavior thus requires extending the range of visuospatial skills under investigation using methods that provide a refined assessment of spatial biases.

The present study aimed to address these issues by testing the effects of anodal tDCS over the left or right parietal cortex of healthy participants while they performed two computerized

tasks, namely a visual search task and a visual imagery task that both precisely measure core aspects of spatial attention. These tasks were chosen because they have remained largely unexplored so far and because they tap on key spatial processes that are essential in everyday life, for example, to maintain an accurate representation of objects in our mind or to allow hand-object interactions in peripersonal space [40]. As they are the tools through which neglect is detected and quantified, they should be considered specific targets for neurorehabilitation. In the present study, the visual search tasks required participants to cross out, on a digital tablet and within a strict time limit, as many targets as possible while ignoring distractors. Under these conditions, cancellation tasks typically reveal, in young healthy adults, a leftward bias, with the first target being most often cancelled on the left side of the template [41–44]. This generally results in a larger number of targets cancelled on the left than on the right side [43, 45]. This mild asymmetry was primarily attributed to right hemispheric dominance [46, 47], offering a valuable model for the inter-hemispheric inhibitory mechanisms involved in neglect after stroke [48–50]. However, it is important to note that an influence of reading habits on this leftward bias has also been reported. Left-to-right readers (Italians) tended to start more often on the left side, whereas right-to-left readers (Israelis) exhibited no bias or a slight rightward bias. The observation that the right bias in right-to-left readers was less pronounced than the leftward bias in left-to-right readers, rather than simply mirroring it, suggests that attention biases in visual search likely results from an interaction between reading habits that may vary due to culture and neurobiological asymmetries that invariably orient attention to the left side of space [51]. Here, we looked at visuo-spatial asymmetries in healthy individuals as a model to predict the counteractive effect of parietal tDCS on neglect [52]. We first computed the average position of detected targets (Center of Cancellation [CoC]) to obtain a sensitive measure indexing not only the number of detected targets but also their spatial distribution. We then performed secondary analyses on the position of the first cancelled target in order to account for the participants' initial bias in the assessment of visuo-spatial asymmetries. These measures are widely used and particularly sensitive for the clinical assessment of neglect in patients [41, 53, 54], especially under time pressure [55]. We also analyzed response accuracy (*i.e.*, the number of targets cancelled on the total number of targets) because previous studies have suggested that tDCS may affect overall performance independently of attention orientation [17, 18, 20]. The visual imagery task required participants to judge whether two shapes presented sequentially are the same or not. The shapes moved through a vertical slot so that participants could only see one section at a time and had to mentally reconstruct the shapes to make their judgement (for a similar task see [34, 56, 57]). No leftward bias has been reported in healthy participants performing this task [34, 35, 57], but the dynamic display ensured sufficient sensitivity and the laterality quotient (LQ) allowed precise measurement of left-right asymmetries in performance. Healthy participants, with left-to-right reading habits, performed the visual search and visual imagery tasks, under both active and sham conditions, with the anode placed over the left or right parietal cortex and the cathode placed over the contralateral orbitofrontal cortex. In the visual search tasks, we expected the first target to be cancelled on the left side of the template, as well as a higher number of detected targets on the left side, indexed by a negative CoC. Assuming that visuo-spatial asymmetries in healthy participants involve–at least partially–a right-hemispheric dominance, the excitatory effect of anodal stimulation over the left or right parietal cortex should lead respectively to a decrease or increase of this leftward bias compared to sham. In the visual imagery task, we expected tDCS to selectively improve the discrimination of shapes that differ on the side contralateral to the anode.

## 2. Materials and methods

### 2.1. Participants

Forty-eight undergraduate students (33 women and 15 men; mean age ± standard deviation: 23 ± 4 years), with left-to-right reading habits, participated in this experiment between June 19, 2015, and August 9, 2017. The sample size was defined on a sample size analysis performed in G*Power indicating that 46 participants were required to detect an interaction of average effect size (Cohen's $f$ = 0.25) between tDCS condition (2 levels) and hemisphere (2 levels) with a high power criterion (0.9) in a mixed model. It is worth noting that the effect size calculation was based on a medium effect size, as we sought for a non-negligible, useful, and theoretically meaningful effect. In particular, we were interested in effects sufficiently large to motivate the use of this protocol in brain-lesioned patients. All participants reported themselves as right-handed and having a normal or corrected-to-normal vision. They had no history of neurological disorders and were unaware of the study's purpose. All participants provided written informed consent prior to the experiment, which was approved by the local Ethics Committee (Comité d'Ethique Hospitalo-Facultaire Saint-Luc UCL; Registration number: B40320108544).

### 2.2. Task and procedure

The left parietal cortex was stimulated in half of the participants and the right parietal cortex was stimulated in the other half. Each participant completed two sessions scheduled on different days. The only difference between the two sessions concerned the setting of the parameters, which were unknown to the participants and allowed for either active or sham stimulation (see next section). In each session, participants performed a visual search task composed of three different cancellation tests followed by a visual imagery task (referred below as the *Cloud* task). The cancellation tests were run on a Dell Latitude XT3 laptop with a 13-inch pivotable touchscreen (29.4 x 16.6 cm; resolution: 1024 x 768 pixels). The center of the screen was aligned with the participant's midline and the touchscreen was oriented horizontally so that the display faced upward (eye-screen distance: 50 cm). The tests used different targets and distractors: (1) the *Mesulam Cancellation Test* [44] included 60 sun drawings (☼) interspersed with a variety of 300 foils; (2) the *Circle Discriminative Cancellation (Ota) Test* [58] included 20 full circles interspersed with 40 pseudo-circles with a missing portion on the right or on the left; and (3) the *Star Cancellation Test* [59] included 56 small stars interspersed with 76 foils consisting of larger stars, letters and short words. Each cancellation test was performed twice in succession within a session. The second presentation consisted in a left-to-right flipped version of the test. In all tests, half of the targets were situated on the left side of the screen and the other half on the right side. The participants were asked to cross as many targets as possible using a digitizing pen with their right dominant hand. The marks made by the pen remained visible during task performance and they were saved in an output image that was used for subsequent analyses, closely mimicking the stroke of a pencil in paper and pencil tasks (Fig 1). After 30 seconds, the display disappeared, and the experimenter started the next test after reminding the instructions to the participants. The order of the three cancellation tests was counter-balanced between participants and kept consistent across active and sham tDCS sessions. Each cancellation test was preceded by a practice test that required crossing short lines displayed randomly all over the screen within 30 seconds [60]. This practice test allowed participants to get familiar with the material.

The *Cloud Task* consisted of a computerized task adapted from Ogden [56]. Two cloud-like shapes of 4.5 cm wide and 2.5 cm high moved one after the other through a vertical slit of 1.4

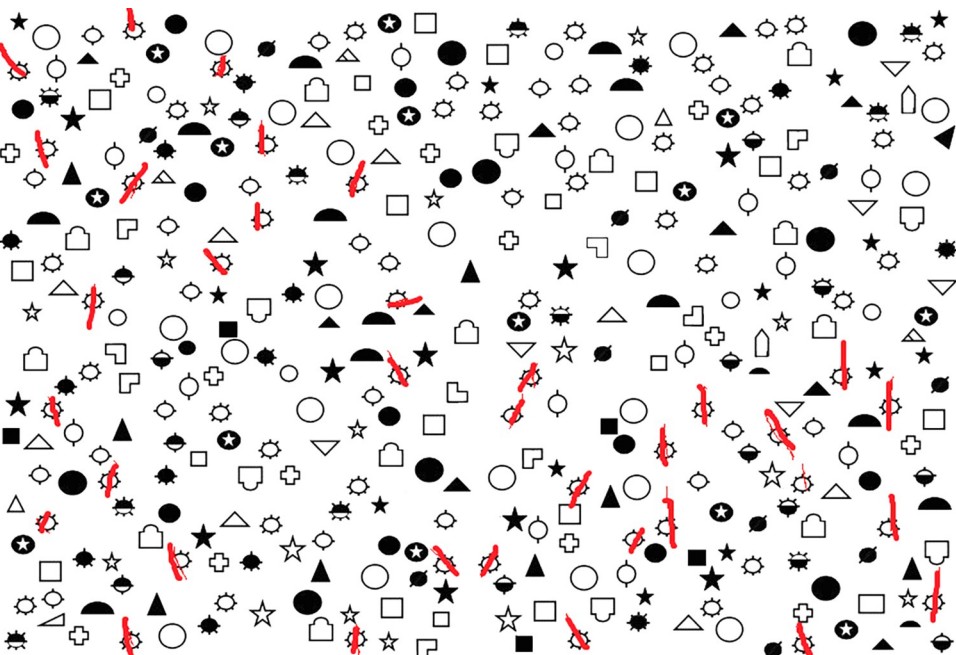

**Fig 1. Performance of one representative participant in the Mesulam cancellation test [44].** Red marks indicate all the stimuli cancelled by the participant within the 30 seconds allowed to do the task.

cm allowing participants to view only one section of each shape at a time. Each shape took 2 seconds to pass through the slot with a blank interval of 2 seconds between the two shapes. The stimuli included 6 pairs of shapes that differed on the left side, 6 pairs that differed on the right side, and 12 identical pairs. The participants were asked to decide whether the two shapes were identical or different by pressing the keys corresponding to the upward and downward arrow, respectively, using their right hand. All pairs were presented from left to right and from right to left in two separate blocks. The order of the two blocks was systematically counterbalanced between participants and the pairs were randomly intermixed within each block. In order to minimize the use of verbalization strategies, such as associating the cloud shape with an object name, participants were asked to repeat the syllable "ba", during each trial, at a 1 Hz pace (trained with a metronome before the experiment). The Cloud Task was performed after the cancellation tests.

## 2.3. Transcranial direct current stimulation

The stimulated hemisphere (left *vs*. right) varied between participants: the anode (5 x 7 cm) was placed over the left posterior parietal cortex (P3) in half of the participants and over the right posterior parietal cortex (P4) in the other half of the participants (according to the international 10–20 electrode placement system). The cathode (5x7 cm) was placed over the orbitofrontal cortex of the hemisphere contralateral to the anode. The stimulation (active *vs*. sham) varied within participants: the active and sham conditions were tested on two different days with a 48-hour interval to avoid spillover effects. On the participant's arrival, the general procedure as well as the functioning of the tDCS was explained to them. The experimenters then positioned the electrodes, encased in a saline-soaked sponge, over the target areas. After participants had received the instructions, tDCS was delivered through a battery-driven electrical stimulator (DC Stimulator Plus, NeuroConn, Ilmenau, Germany). In the active condition, a

constant current of 1.5 mA was delivered for 20 minutes. In the sham condition, the stimulator was programmed with the same parameters except that the current was turned off after 30 seconds. In both conditions, the stimulation was applied with a 30-second fade-in and fade-out phase. Previous studies reported no adverse effects with these parameters [61]. The participants started performing the tasks after a fixed delay of 2 minutes after stimulation onset and completed them within the 20 minutes of stimulation. The four possible orders resulting from the possible combinations of hemisphere (left and right) and tDCS (active and sham) conditions, i.e., (1) left tDCS on day 1 and left sham on day 2, (2) left sham on day 1 and left tDCS on day 2, (3) right tDCS on day 1 and right sham on day 2, and (4) right sham on day 1 and right tDCS on day, were systematically counterbalanced across the 48 participants.

## 2.4. Data analysis

All data were analyzed using mixed-effects models implemented with the *lme4* R package [62]. The appropriate model for each dependent variable was selected based on its distribution, which was assessed by computing skewness and visualizing histograms. Model parameters were estimated using the Laplace approximation and statistical significance was evaluated using Wald's $\chi^2$ test, which quantifies the proportion of variance explained by the fixed effects [63]. Bonferroni-corrected post-hoc pairwise contrasts were performed using the *emmeans* R package [64] and effect sizes (measured as Cohen's *d*) were computed by taking the difference between the means divided by the square root of the variance of the intercept of the participants [65]. While the primary endpoint of the investigation of visual search performance was the spatial distribution of cancelled targets (i.e., CoC), we further explored the effects of tDCS on the starting position (i.e., first cancelled target) to account for the participants' initial bias and extend the assessment of left-right asymmetries to other clinically relevant measures.

**2.4.1 Cancellation tests.** *2.4.1.1. Position of the first cancelled target.* To analyze the starting position, we extracted the mean x-coordinate of the first mark performed at each trial. The distribution of these coordinates was significantly right-skewed as compared to the normal distribution (skewness ± SE: 0.52 ± 0.10), $z = 5.14$, $p < .001$. We thus investigated the effect of tDCS on the starting position by fitting a Gamma GLMM for right-skewed distributions with TEST, TDCS AND HEMISPHERE as fixed effects and participant as random intercept. To enhance comprehension, the estimated means were expressed relative to the x-coordinate of the screen's center, with negative values indicating a leftward position and positive values indicating a rightward position.

*2.4.1.2. Centre of cancellation.* To analyze the spatial distribution of the targets cancelled, we calculated the individual CoC values by summing the horizontal positions of the targets cancelled by the participant and by dividing this sum by the total number of targets cancelled. The result was normalized so that participants who cancelled all targets would receive a score of 0, while participants who only cancelled the leftmost items would receive a score of -1 and participants who cancelled only the rightmost items would receive a score of +1. Cancellation tests in healthy young adults with left-to-right reading habits typically give rise to negative CoC values (most targets crossed are to the left of the center). This measure has the advantage to reflect the spatial distribution of the targets cancelled while taking into account the number of targets cancelled [53]. The distribution of the CoC was unskewed as compared to the normal distribution (-0.09 ± 0.10), $z = 0.85$, $p = .393$. We thus investigated the effect of tDCS on the spatial distribution of the cancelled targets by fitting a Linear Mixed Model (LMM) on the CoC values with TEST, TDCS and HEMISPHERE as fixed effects and participant as random intercept.

*2.4.1.3. Overall accuracy.* To analyze overall accuracy in visual search, we computed the percentage of targets cancelled in each test. The distribution of these percentages was left-skewed

as compared to the normal distribution (-0.47 ± 0.10), $z = 4.58$, $p < .001$. We thus inverted the skew by adding a negative sign to the percentage values. We then added a constant (i.e., 2) to avoid negative values and fitted a Gamma GLMM (see [66] for similar procedure). All estimates were retransformed to the original scale before being reported.

*2.4.1.4. Test-retest reliability*. We calculated Pearson correlation coefficients to evaluate the consistency of each dependent variable between the normal (first) and flipped (second) versions of each test during the sham session.

**2.4.2 Cloud task.** *2.4.2.1. Laterality quotient*. To investigate the effect of tDCS on left-right asymmetries in the Cloud Task, we computed the laterality quotient (LQ) for each block by subtracting the number of correct responses for pairs of clouds that differed on the left side ($CR_L$) from the number of correct responses for pairs of clouds that differed on the right side ($CR_R$), dividing this value by the total number of correct responses, and multiplying the result by 100. A negative LQ indicated better performance for pairs that differed on the left side, while positive values indicated better performance for pairs that differed on the right side [56]. The distribution of the LQ values was unskewed as compared to the normal distribution (-0.26 ± 0.17), $z = 1.50$, $p = .133$. We thus entered the LQ values in a LMM with TEST, TDCS and HEMISPHERE as fixed effects and participant as random intercept. We also computed a correlation between the LQ of the sham session and the CoC in each of the cancellation tests of the sham sessions to test whether there was an association between the bias observed in visual search and visual imagery.

*2.4.2.2. Overall accuracy*. Finally, to test whether tDCS affects overall accuracy in visual imagery, we ran a binomial GLMM on the accuracy at each trial with SIMILARITY (left difference, right difference *vs*. identical shapes), TDCS (sham *vs*. active) and HEMISPHERE (left *vs*. right anodal stimulation) as fixed factors and participant as random intercept.

# 3. Results

## 3.1. Visual search

### 3.1.1 Position of the first cancelled target.

The mean x-coordinate estimated by the GLMM in each condition is reported in Table 1. The model revealed a significant main effect of TEST, $\chi^2(2) = 9.59$, $p = .008$. The 95% confidence intervals for the estimated x-coordinate of the first mark in each test indicated it was significantly lower than 0, meaning it was leftwards, in all three cancellation tests: Mesulam (estimated mean ± SE: -88 ± 31 pixels, corresponding to 2.53 ± 0.95 cm), 95% CI = [−145, −31], Star (-90 ± 31 pixels or 2.58 ± 0.95 cm), 95% CI = [−

**Table 1. Means ± S.E. of each dependent variable in the cancellation tests as estimated by the respective models, expressed as a function of the test, stimulated hemisphere (anodal) and tDCS condition.** Pixel origin for the position of the first cancelled target was the center of the screen.

| | | Left Hemisphere | | Right Hemisphere | |
|---|---|---|---|---|---|
| | | **Sham** | **tDCS** | **Sham** | **tDCS** |
| **Position of first cancelled target (X coordinates, in pixels)** | Mesulam | -67 ± 28 | -1 ± 28 | -103 ± 50 | -181 ± 51 |
| | Ota | -87 ± 31 | -76 ± 29 | -169 ± 51 | -198 ± 49 |
| | Star | -51 ± 35 | 46 ± 37 | -190 ± 50 | -166 ± 51 |
| **CoC (normalized values)** | Mesulam | -0.13 ± 0.04 | -0.13 ± 0.04 | -0.07 ± 0.04 | -0.08 ± 0.04 |
| | Ota | -0.03 ± 0.04 | -0.06 ± 0.04 | -0.03 ± 0.04 | -0.02 ± 0.04 |
| | Star | -0.06 ± 0.04 | -0.06 ± 0.04 | -0.04 ± 0.04 | -0.04 ± 0.04 |
| **Percentage of cancelled targets (%)** | Mesulam | 50 ± 3 | 50 ± 3 | 51 ± 3 | 52 ± 3 |
| | Ota | 82 ± 3 | 79 ± 3 | 81 ± 3 | 86 ± 3 |
| | Star | 83 ± 3 | 82 ± 3 | 82 ± 3 | 83 ± 3 |

151, −29]) and Ota (-133 ± 29 pixels or 3.82 ± 0.83 cm), 95% CI = [−193, −72]. Post-hoc pair-wise contrasts further showed that the first mark was significantly more leftwards for the Ota test compared to the Mesulam test, $z$ = -3.58, $p$ = .001, $d$ = 0.25, and the Star test, $z$ = -3.23, $p$ = .003, $d$ = 0.25, while there was no significant difference between the Mesulam and Star tests, $z$ = 0.14, $p$ = 1.000, $d$ = 0.01.

There was a significant interaction between TDCS and HEMISPHERE, $\chi^2(1)$ = 6.75, $p$ = .009, between TDCS and TEST, $\chi^2(2)$ = 10.80, $p$ = .004, and between TEST and HEMISPHERE, $\chi^2(2)$ = 7.84, $p$ = .019. These two-way interactions were embedded in a significant three-way interaction between TDCS, HEMISPHERE and TEST, $\chi^2(2)$ = 8.88, $p$ = .012. Post-hoc pairwise contrasts between active and sham tDCS for the Mesulam test showed a significant rightward displacement of the first cancelled target of 66 pixels (1.89 cm) in the group receiving left anodal stimulation, $z$ = -3.23, $p$ = .001, $d$ = 0.38, and a significant leftward displacement of 78 pixels (2.24 cm) in the group receiving right anodal stimulation, $z$ = 2.82, $p$ = .005, $d$ = 0.45. In the Star test, there was a significant rightward displacement of 97 pixels (2.78 cm) in the active as compared to the sham condition in the group receiving left anodal stimulation, $z$ = 3.47, $p$ < .001, $d$ = 0.56, while no significant difference was observed in the group receiving right anodal stimulation, $z$ = 1.25, $p$ = .211, $d$ = 0.14. Finally, in the Ota test, no significant differences were observed, whether in the left, $z$ = 0.61, $p$ = .583, $d$ = .06, or right, $z$ = -1.63, $p$ = .103, $d$ = 0.17, hemisphere group (Fig 2).

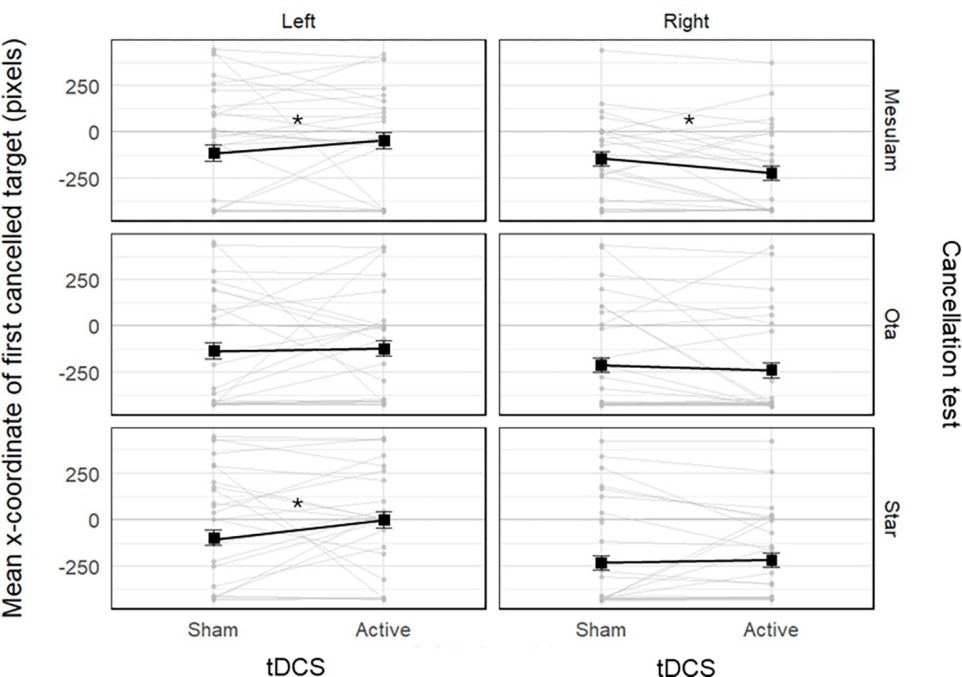

**Fig 2. Stripcharts depicting the mean x-coordinate of the first cancelled target, as a function of tDCS condition (sham vs. active), stimulated hemisphere (left vs. right anodal), and test (Mesulam, Ota vs. Star).** Each of the 6 plots includes data from 24 participants. The black connected squares represent the estimated means derived from a Gamma GLMM for both sham and active tDCS conditions, with error bars indicating the SE. The grey connected dots represent the observed mean values for each participant in both the sham and active tDCS condition. The zero value corresponds to the center of the template, while negative and positive values indicate biases towards the left and right sides of the template, respectively. Asterisks represent significant differences between the sham and active tDCS conditions. In the Mesulam test, anodal stimulation of the left and right parietal cortex shifted the position of the first mark in the contralateral direction compared to sham stimulation. In the Star test, there was a contralateral bias for the anodal stimulation of the left but not right parietal cortex. No significant difference was observed in the Ota test.

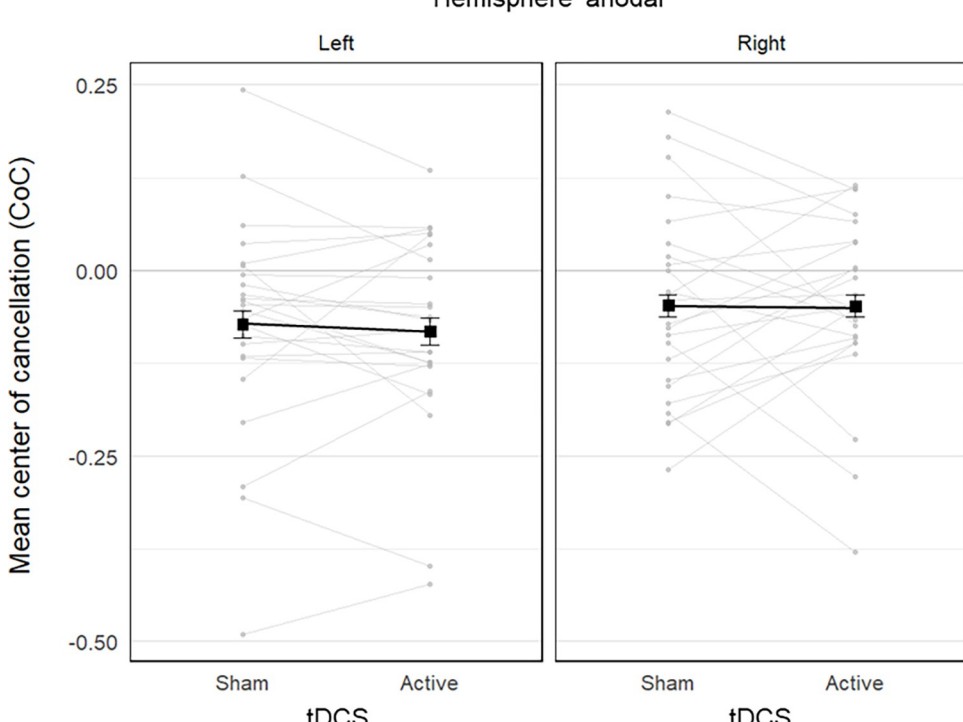

**Fig 3. Stripcharts of the centers of cancellation (CoC) in the cancellation tests as a function of tDCS condition (sham vs. active), and stimulated hemisphere (left vs. right anodal).** Each plot includes data from 24 participants. The black connected squares represent the estimated means derived from a LMM for the sham and active tDCS condition, with error bars indicating the SE. The grey connected dots represent the observed mean values for each participant in both sham and active tDCS conditions. The zero value corresponds to an absence of bias, while negative and positive values indicate biases towards the left and right sides of the template, respectively.

**3.1.2 Center of cancellation.** The mean CoC estimated by the LMM in each condition is reported in Table 1. The model showed a significant main effect of TEST, $\chi^2(2) = 8.54$, $p = .014$. The 95% confidence intervals around the estimated CoC showed that it was significantly lower than 0, meaning it was biased towards the left, in the Mesulam test (-0.10 ± 0.02), 95% CI = [-0.15, -0.06], and in the Star test (-0.05 ± 0.02), 95% CI = [-0.10, -0.01], but not in the Ota test (-0.03 ± 0.02), 95% CI = [-0.08, 0.01]. Post-hoc pairwise contrasts further showed that the CoC was more leftwards in the Mesulam compared to the Ota test, $z = -2.77$, $p = .017$, $d = 0.68$, while there was no significant difference with the Star test, $z = -2.17$, $p = .088$, $d = 0.54$, or between the Ota and Star tests, $z = 0.57$, $p = 1.000$, $d = 0.14$.

The main effect of TDCS and its interaction with HEMISPHERE or TEST were not significant (all p-values > .389, see statistical details in S1 Appendix), suggesting that the CoC was not affected by parietal tDCS. Fig 3 shows that a high proportion of participants was biased toward the left (vs. right side) similarly in the sham (34 vs. 14 participants) and active tDCS condition (32 vs. 16 participants), suggesting that anodal stimulation of the left or right parietal cortex did not modulate the bias observed in the sham condition.

**3.1.3 Percentage of cancelled targets.** The estimated percentage of cancelled targets in each condition is reported in Table 1. The Gamma GLMM showed a significant effect of TEST, $\chi^2(2) = 1045.70$, $p < .001$, indicating that participants cancelled a smaller percentage of targets in the Mesulam test (51 ± 2%), than in the Star test (83 ± 2%), $z = -28.84$, $p < .001$, $d = 5.18$,

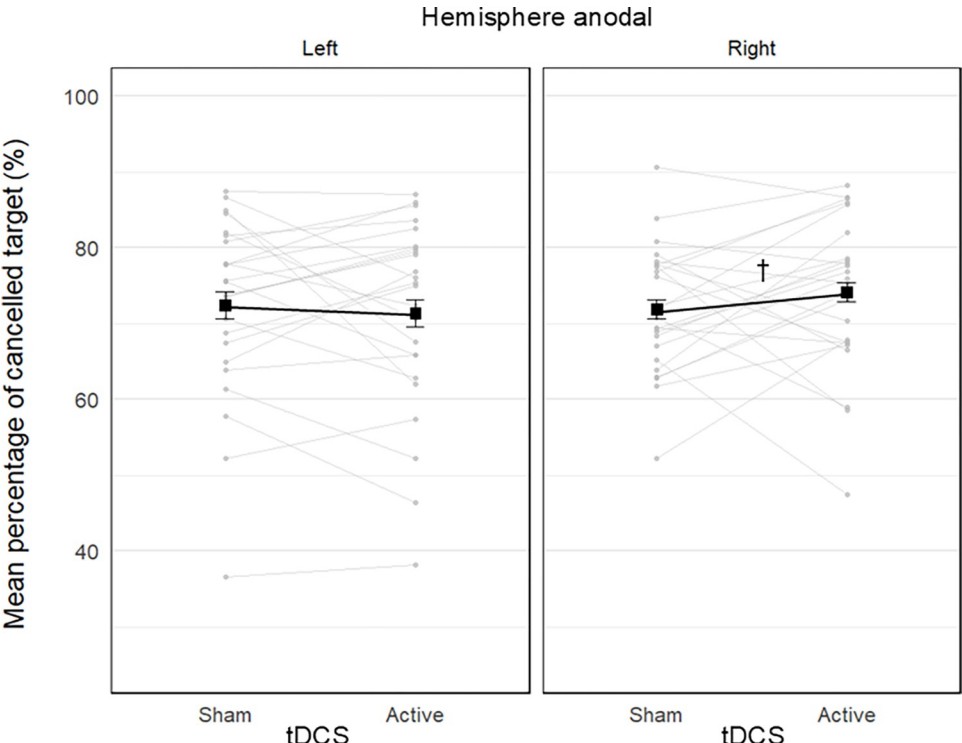

**Fig 4. Stripcharts of the mean percentage of targets cancelled in the cancellation tests as a function of tDCS condition (sham vs. active), and stimulated hemisphere (left vs. right anodal).** Each plot includes data from 24 participants. The black connected squares represent the estimated means derived from a Gamma GLMM for the sham and active tDCS condition, with error bars indicating the SE. The grey connected dots represent the observed mean values for each participant in both sham and active tDCS conditions. The dagger symbol represents a marginal difference between the sham and active tDCS conditions. Anodal stimulation of right parietal cortex marginally improved overall accuracy compared to sham stimulation.

and in the Ota test (83 ± 2%), $z = -29.21$, $p < .001$, $d = 5.11$, while there was no significant difference between the Star and Ota tests, $z = 0.43$, $p = 1.000$, $d = 0.07$.

There was also a significant TDCS by HEMISPHERE interaction, $\chi^2(2) = 4.81$, $p = .028$, suggesting the effect of TDCS differed between the left and right hemisphere groups. Post-hoc pairwise contrasts showed a marginal improvement in the active (74 ± 3%) compared with the sham (71 ± 3%) condition in the right hemisphere group, $z = 1.82$, $p = .069$, $d = 0.35$, while no significant difference was observed for the left hemisphere group (active: 70 ± 3%; sham: 72 ± 3%), $z = -0.86$, $p = .388$, $d = 0.17$ (Fig 4). All other effects of the model were not significant (all $p$-values $> .164$; see details in S2 Appendix).

**3.1.4 Test-retest reliability.** Regarding the x position of the first cancelled target, we observed a significant positive correlation between the normal (first) and flipped (second) version for the Mesulam, $r = .38$, $p = .007$, Ota: $r = .72$, $p < .001$, and Star, $r = .72$, $p < .001$, tests. Regarding the CoC, a significant correlation was found only for the Star test, $r = .37$, $p = .011$, whereas no significant correlation was observed for the Mesulam, $r = .01$, $p = .948$, and Ota, $r = -.18$, $p = .211$, tests. Finally, for the percentage of cancelled targets, positive correlations were evident for the Mesulam, $r = .63$, $p < .001$, Ota, $r = .70$, $p < .001$, and Star, $r = .60$, $p < .001$, tests.

## 3.2. Visual imagery

**3.2.1. Laterality quotient.** The estimated LQ in each condition is reported in Table 2. The LMM on the LQs showed no significant effect of TDCS, $\chi^2(1) = 0.27$, $p = .603$, HEMISPHERE, $\chi^2(1)$

**Table 2. Means ± S.E. of each dependent variable in the Cloud task as estimated by the respective models, expressed as a function of similarity (for overall accuracy only), stimulated hemisphere and tDCS condition.**

|  | Left Hemisphere | | Right Hemisphere | |
|---|---|---|---|---|
|  | **Sham** | **tDCS** | **Sham** | **tDCS** |
| **Laterality Quotient** | -0.07 ± 4.65 | 2.56 ± 4.65 | 1.6 ± 4.65 | -5.87 ± 4.70 |
| **Overall Accuracy (%)** |  |  |  |  |
| • Left difference | 68 ± 1 | 68 ± 1 | 63 ± 1 | 67 ± 1 |
| • Right difference | 68 ± 1 | 70 ± 1 | 65 ± 1 | 62 ± 1 |
| • Identical | 84 ± 1 | 84 ± 1 | 85 ± 1 | 82 ± 1 |

= 0.51, $p$ = .475, or TDCS by HEMISPHERE interaction, $\chi^2(1)$ = 1.19, $p$ = .275 (Fig 5). The 95% confidence intervals showed that the estimated LQ was not different than 0, neither in the sham (0.79 ± 3.28), 95% CI = [-5.70, 7.29], nor in the active (-1.65 ± 3.30), 95% CI = [-8.18, 4.88], tDCS conditions.

None of the correlations between the LQ of the Cloud task and the CoC of the cancellation tests were statistically significant. The Pearson correlation coefficients and associated $p$-values are reported in Table 3.

**3.2.2. Overall accuracy.** The mean accuracy estimated by the binomial GLMM in each condition is reported in Table 2. The model showed a significant effect of SIMILARITY, $\chi^2(2)$ = 177.79, $p$ < .001, indicating that identical shapes (estimated mean + S.E.: 83.6 ± 0.7%) were

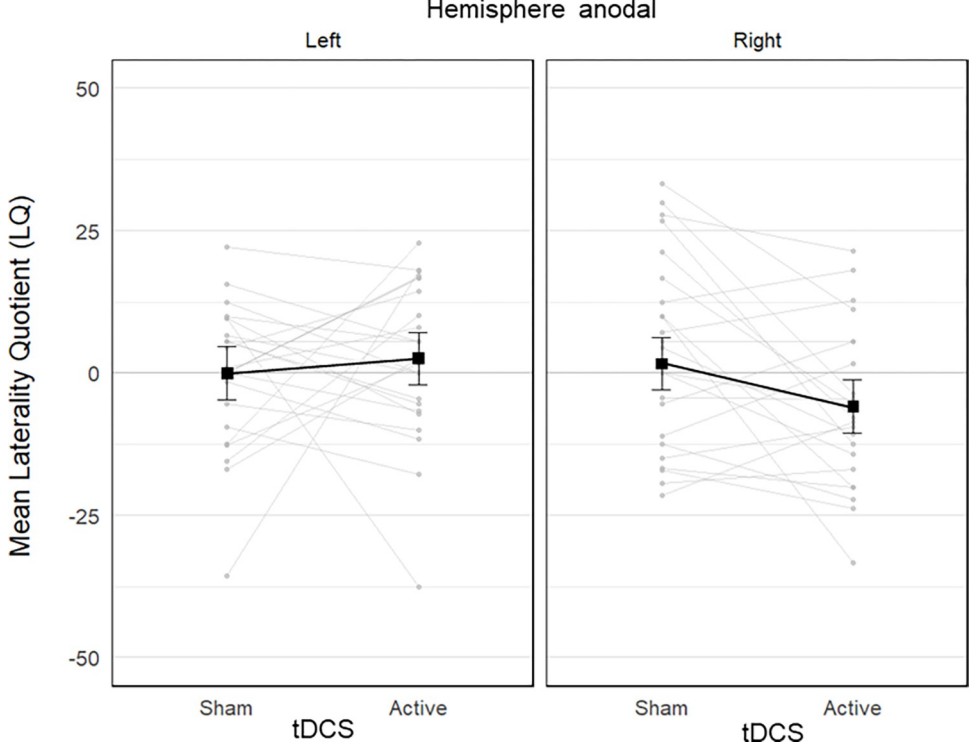

**Fig 5. Stripchart of the mean laterality quotient (LQ) in the Cloud task as a function of tDCS condition (sham vs. active), and stimulated hemisphere (left vs. right anodal).** Each plot includes data from 24 participants. The black connected squares represent the estimated means derived from LMM for both sham and active tDCS conditions, with error bars indicating the SE. The grey connected dots represent the observed mean values for each participant in both sham and active tDCS conditions.

**Table 3. Pearson r coefficients and associated p-values of the correlation between the LQ of the Cloud task and the Center of Cancellation (CoC) of each cancellation test.**

|  | Mesulam (CoC) | | Ota (CoC) | | Star (CoC) | |
|---|---|---|---|---|---|---|
|  | r | p | r | p | r | p |
| Cloud Task (LQ) | .12 | .403 | .06 | .705 | -.03 | .846 |

recognized more accurately than shapes differing on the left (66.7 ± 1.1%), $z = 11.41$, $p < .001$, $d = 2.35$, or right side (66.2 ± 1.2%), $z = 11.21$, $p < .001$, d = 2.31, while there was no significant difference between shapes differing on the left and those differing on the right, $z = 0.19$, $p = 1.00$, $d = 0.04$ (Fig 6). Other main effects and interactions were not significant (all $p$-values > .301; see detailed statistics in S3 Appendix).

## 4. Discussion

The goal of this study was to investigate the modulatory effects of parietal tDCS in healthy participants performing visual search and visual imagery tasks, which have remained largely unexplored despite their ability to reveal attention biases after stroke, especially in the neglect syndrome. We hypothesized that the excitatory effect of anodal stimulation would alter the balance of inter-hemispheric activity, biasing attention to the contralateral side of both physical and mental spaces, as assessed by the visual search and visual imagery tasks, respectively. Specifically, we anticipated the typical leftward bias observed in visual search to be amplified

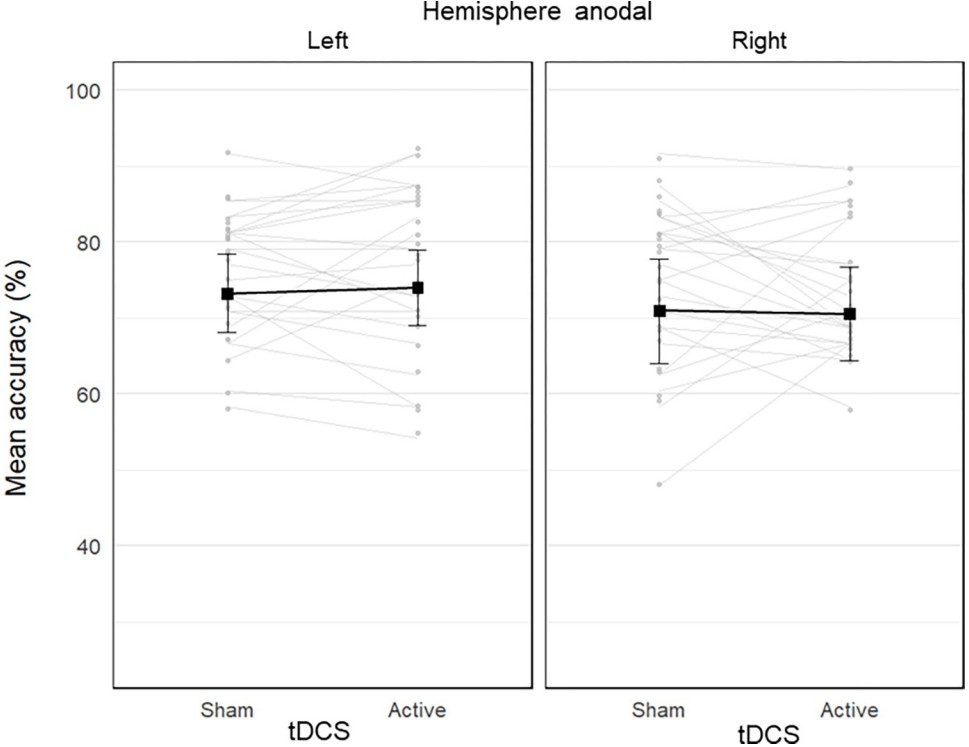

**Fig 6. Stripcharts of the mean accuracy in the Cloud task as a function of tDCS condition (sham vs. active), and stimulated hemisphere (left vs. right anodal).** Each plot includes data from 24 participants. The black connected squares represent the estimated probabilities derived from a binomial GLMM for both sham and active tDCS conditions, with error bars indicating the SE. The grey connected dots represent the observed mean values for each participant in both sham and active tDCS conditions.

by right anodal stimulation and reduced by left anodal stimulation. Additionally, we considered the possibility of tDCS effects on overall test accuracy, regardless of hemifield, as some studies have reported general enhancements in attention [19, 20]. Although the visual imagery task typically does not reveal a leftward bias in healthy individuals [34, 56, 57], we expected tDCS to selectively improve the discrimination of shapes that differ on the side contralateral to the anode.

In the visual search tasks, performance in the control sham condition was characterized by a typical leftward bias in the selection of the first target and in the spatial distribution of all cancelled targets, as evidenced by the x-coordinate of the first cancellation mark and the CoC. As previously suggested, participants might have selected the first target on the left side of the template because of their reading habits [51, 67]. However, we found that parietal anodal tDCS induced a contralateral shift in the selection of the first cancelled target by over 60 pixels, thus more than 1.7 cm, with the initial leftward bias observed during the sham session being amplified (i.e., more lateral) during right anodal stimulation and reduced (i.e., more central) during left anodal stimulation. The study was thus successful in demonstrating the efficiency of parietal tDCS in modulating spatial biases in healthy participants. This indicates that reading habits do not override biologically-rooted asymmetries in the initial selection bias since it can be shifted in either direction depending on which hemisphere is being stimulated (i.e., left vs. right anodal), as predicted by the inter-hemispheric rivalry hypothesis.

Three results add nuance to the conclusion that parietal tDCS can be used to modulate spatial attention in visual search. First, the three cancellation tests showed a different sensitivity to tDCS effects. In the Mesulam test, both left and right hemisphere anodal stimulation induced a contralateral shift, whereas in the Star cancellation test, only right anodal stimulation resulted in a significant leftward displacement. In the Ota test, no modulation of performance was observed. The Mesulam (60 targets among 300 foils) and Star (56 targets among 76 foils mixing shapes and letters) tests count a greater number or diversity of distractors than the Ota test (20 full circles among 40 open circles). This suggests parietal modulation may be more evident in demanding contexts where attention is divided between several competing stimuli, as previously observed in visual detection tasks [19, 68]. Second, the counteracting effect of parietal tDCS on the initial leftward bias was not reflected in the overall spatial distribution of cancelled targets. Contrary to what was observed for the first cancelled target, tDCS did not alter the spatial distribution of other targets cancelled. The overall spatial distribution remained biased towards the left side of the template, as evidenced by negative CoC values. One might suggest that the left bias persisted because participants started on the left side and were unable to reach the right side of the template within the 30-seconds time limit. While the spatial distribution of cancelled targets is indeed typically anchored to the first cancelled target, this explanation is unlikely, as participants typically crossed the midline of the template at an average of 11.7 ± 6.8 seconds after trial onset, leaving enough time to explore both sides of the template. Hence, the lower proportion of targets cancelled on the right side cannot be attributed to a lack of time. Instead, in-flight corrections or idiosyncratic strategies emphasizing top-to-bottom rather than left-to-right exploration may have contributed to blurring the effect of tDCS on subsequent cancellations. A detailed visual inspection of individual performance revealed that participants generally followed a non-linear path occasionally interrupted by loops and revisits (see individual cancellation sheets made available at https://osf.io/4dbs5/). The CoC was also characterized by a lower test-retest reliability, compared to other indices (i.e., first cancelled target, percentage of cancelled targets) that should thus be preferred for the assessment of visuospatial biases in future tDCS studies. Third, it is worth noting that the starting position in visual search tasks was not the primary endpoint of the present study and that other measures of spatial attention in these tasks did not reveal consistent effects of tDCS. The

overall performance in visual search slightly improved during right anodal stimulation, independently of the hemifield, as evidenced by a greater percentage of cancelled targets compared to sham. Although the increase was marginally significant, we chose to report it because it adds to previous evidence that parietal tDCS may also help patients recover a sufficient level of attention to detect changes across the whole visual space [19]. This potential should not be overlooked because several studies have indicated that lateralized and non-lateralized aspects of attention systematically interact in hemineglect [69–71]. The effect of parietal tDCS on visual search also opens up perspectives for the neurorehabilitation of other visual impairments, as to help overcome the blind area in lateral homonymous hemianopia [72, 73]. Further studies are, however; necessary to consolidate the present findings and refine the margin of improvement that can reasonably be expected in brain-lesioned patients receiving anodal stimulation over the parietal cortex in a therapeutic context.

In the visual imagery task, we found no evidence that parietal tDCS has biased attention toward the left or right side of the shape representation maintained in short-term memory. In the sham condition, shapes with left-sided differences were detected as accurately as those with right-sided differences, and anodal stimulation did not modify this pattern of performance. This might indicate that mental imagery skills rely on mechanisms resilient to weak excitatory currents into the posterior parietal cortex. The unity of spatial attention is very debated in the literature [24, 40, 74, 75]. For instance, clinical observations in neglect patients showed a double dissociation between performance on cancellation *vs.* line bisection tests [76–78] and between attention orientation to perceived *vs.* imagined objects [79–81]. These dissociations imply that visual imagery may involve different cognitive processes and brain networks than those involved in tasks found to be affected by tDCS such as visual search or line bisection (for a review, see [82, 83]). Neuropsychological studies suggested that visual search, as assessed through cancellation tests, relies on parietal, frontal and subcortical areas, while line bisection mainly depends on the parietal lobes [84, 85]. Deficits in representational space are often associated with lesions to the temporoparietal cortices, but also frequently with lesions to the occipital and frontal cortices including subcortical structures (for a review, see [86]). Likewise, neuroimaging data from healthy subjects indicated that orienting attention to imagined *vs.* perceived objects selectively activates the prefrontal cortex [87]. In the present study, we found no association between the spatial biases observed in visual search and those observed in visual imagery. Hence, beside a common parietal network controlling spatial attention, the visual imagery tasks might involve additional brain systems compared to the tasks revealing an effect of tDCS on spatial attention [88, 89]. Thus, the reason why parietal tDCS was not sufficient to modify performance in visual imagery may lie in the different network used by this task compared to other visuospatial tasks dealing mainly with the resources of the posterior parietal cortex. Among other possibilities, a large redundant network can help mitigate the effects of parietal tDCS. A limitation of the present study is that the visual imagery task was systematically administered after the visual search task. Although there is no theoretical reason to assume that neuromodulatory effects decreased between the beginning and end of the stimulation [1, 90], we cannot rule out, for example, that prior performance of the visual search task boosted attentional resources, making the visual imagery task more resistant to the effect of tDCS.

Because we assessed spatial attention using two sensitive tasks and a method that proved adequate to reveal spatial biases in healthy and brain-lesioned individuals, we are confident that the results give valid indications about the efficacy of anodal tDCS to modulate spatial attention in the context of clinical tasks. However, these results are linked to the specific parameters used in the present study and testing other parameter combinations is necessary to optimize the proposed tDCS protocol and refine conclusions about which aspect of attention performance may be affected by parietal tDCS. It has been suggested that current strength may

interact with baseline performance [91], with high levels of discrimination sensitivity leading to lateralized biases after low- (1mA) but not high-intensity (2mA) tDCS and the reverse for low levels of discrimination sensitivity. In the absence of independent measures of discrimination sensitivity, we could not address this hypothesis in the present study, but we could refute it based on the former results of a conceptual replication study [75]. While our results converge with those of a previous study to show that 20 minutes of anodal tDCS over the left parietal cortex is sufficient to counteract leftward biases either in visual search or in a grayscale task [15], the positioning of the electrodes deserves further investigation. A common feature of the studies reporting an effect of tDCS on spatial attention is the positioning of the cathode over the right parietal cortex [88, 89]. Recent studies further suggested that bilateral tDCS, combining left anodal and right cathodal stimulation, offers a valid alternative to modulate spatial attention [16, 19, 92, 93], but these conclusions remain to be extended to visual search and visual imagery as studies have mainly focused on visual detection so far. Finally, further research is needed to assess the respective efficiency of on-line (during task performance) and off-line (before task performance) tDCS protocols in modifying the balance of attention across the visual space. In the present study, anodal stimulation was delivered continuously during task performance but started 2 minutes prior to the first task. We did so to anticipate a possible delay in the depolarization of the neural tissue but admittedly this delay could be longer than expected and the observed effects in visual search could actually be enhanced with off-line protocols that temporally dissociate the stimulation and the testing, as in previous studies on visuospatial detection [14].

To conclude, the study tested the effects of parietal tDCS in young, healthy individuals performing visual search and mental imagery using computerized tasks adapted from neuropsychological tests commonly used in clinical practice. In the visual search task containing the most distractors, anodal stimulation of the parietal cortex shifted the x-coordinate of the first examined position in the direction opposite to the stimulated hemisphere, as predicted by the hemispheric rivalry hypothesis. In the mental imagery task, performance remained unaffected, suggesting that visuospatial imagery skills rely on mechanisms resilient to weak excitatory currents into the posterior parietal cortex. These results give important indications for the use of tDCS as a way to correct abnormal activity after stroke and they emphasize the need to couple neurophysiological and cognitive research to develop efficient neurorehabilitation strategies.

## Supporting information

**S1 Appendix. Detailed statistics of the LMM on the CoC in the cancellation tests.**
(DOCX)

**S2 Appendix. Detailed statistics of the Gamma GLMM on the percentage of cancelled targets in the cancellation tests.**
(DOCX)

**S3 Appendix. Detailed statistics of the binomial GLMM on the accuracy at the Cloud task.**
(DOCX)

## Acknowledgments

We thank Samuel Di Luca for his help in setting up the visual search task.

## Author Contributions

**Conceptualization:** Laurie Geers, Valérie Dormal, Mario Bonato, Yves Vandermeeren, Nicolas Masson, Michael Andres.

**Data curation:** Laurie Geers.

**Formal analysis:** Laurie Geers, Nicolas Masson.

**Funding acquisition:** Mario Bonato, Yves Vandermeeren, Michael Andres.

**Investigation:** Laurie Geers, Valérie Dormal, Nicolas Masson.

**Methodology:** Valérie Dormal, Mario Bonato, Yves Vandermeeren, Michael Andres.

**Project administration:** Michael Andres.

**Resources:** Valérie Dormal, Yves Vandermeeren, Michael Andres.

**Software:** Valérie Dormal.

**Supervision:** Laurie Geers, Valérie Dormal, Michael Andres.

**Validation:** Valérie Dormal, Mario Bonato, Yves Vandermeeren, Michael Andres.

**Visualization:** Laurie Geers.

**Writing – original draft:** Laurie Geers.

**Writing – review & editing:** Laurie Geers, Valérie Dormal, Mario Bonato, Nicolas Masson, Michael Andres.

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
