## [Decision Letter · Decision Letter 0]

29 May 2024

PONE-D-24-02796TDCS-resistant pseudoneglect in visual search and imageryPLOS ONE

Dear Dr. Geers,

Thank you for submitting your manuscript to PLOS ONE. After careful consideration, we feel that it has merit but does not fully meet PLOS ONE’s publication criteria as it currently stands. Therefore, we invite you to submit a revised version of the manuscript that addresses the points raised during the review process.

As you will see reviewer 2 raised several critical points regarding the appropiateness of the methodology used in the current paper, which I invite you to consider carefully one by one.

We look forward to receiving your revised manuscript.

Kind regards,

Luigi Cattaneo, MD, PhD

Academic Editor

PLOS ONE

Journal Requirements:

 [The research of MA was supported by grants T.0245.16 and T.0047.18 from the FRS-FNRS and by the Fonds Spécial de Recherche of the Université catholique de Louvain (FSR, UCLouvain, Belgium). MB contributed thanks to a grant from MIUR (Dipartimenti di Eccellenza DM 11/05/2017 n. 262). The work of YV was supported by grant 1.R.506.16 from the Fonds National de la Recherche Scientifique (FRS-FNRS, Belgium), grant 3.4.525.08.F from the Fonds de la Recherche Scientifique Médicale (FRSM, Belgium), and grants from the Fondation Van Goethem-Brichant and Fondation Mont-Godinne.].  

[LG is a scientific collaborator and MA is a research associate at the Fonds National de la Recherche Scientifique (FRS-FNRS, Belgium). The research of MA was supported by grants T.0245.16 and T.0047.18 from the FRS-FNRS and by the Fonds Spécial de Recherche of the Université catholique de Louvain (FSR, UCLouvain, Belgium). MB contributed thanks to a grant from MIUR (Dipartimenti di Eccellenza DM 11/05/2017 n. 262). The work of YV was supported by grant 1.R.506.16 from the Fonds National de la Recherche Scientifique (FRS-FNRS, Belgium), grant 3.4.525.08.F from the Fonds de la Recherche Scientifique Médicale (FRSM, Belgium), and grants from the Fondation Van Goethem-Brichant and Fondation Mont-Godinne. We thank Samuel Di Luca for his help in setting up the visual search task.]

 [The research of MA was supported by grants T.0245.16 and T.0047.18 from the FRS-FNRS and by the Fonds Spécial de Recherche of the Université catholique de Louvain (FSR, UCLouvain, Belgium). MB contributed thanks to a grant from MIUR (Dipartimenti di Eccellenza DM 11/05/2017 n. 262). The work of YV was supported by grant 1.R.506.16 from the Fonds National de la Recherche Scientifique (FRS-FNRS, Belgium), grant 3.4.525.08.F from the Fonds de la Recherche Scientifique Médicale (FRSM, Belgium), and grants from the Fondation Van Goethem-Brichant and Fondation Mont-Godinne.].  

Reviewers' comments:

Reviewer's Responses to Questions

**Comments to the Author**

1. Is the manuscript technically sound, and do the data support the conclusions?

Reviewer #1: Yes

Reviewer #2: Partly

2. Has the statistical analysis been performed appropriately and rigorously? 

Reviewer #1: Yes

Reviewer #2: No

3. Have the authors made all data underlying the findings in their manuscript fully available?

Reviewer #1: Yes

Reviewer #2: No

4. Is the manuscript presented in an intelligible fashion and written in standard English?

Reviewer #1: Yes

Reviewer #2: Yes

5. Review Comments to the Author

Reviewer #1: In two sessions, either tDCS or sham stimulation was applied in healthy participants, on the left hemisphere (n = 23) or right hemisphere (n = 23). Effects of the stimulation on visual search and visual imagery was evaluated, specifically focusing on the visuospatial asymmetry in performance. Anodal tDCS was applied over the parietal cortex, increasing the spontaneous firing rate, assumed to improve visual detection in the contralateral hemifield. The study is relevant to inform treatments of neglect, and conducted properly. I do have some suggestions that the authors may want to take into account.

The authors mention that the cancellation tasks are sensitive to pseudoneglect, especially since a time limit was used. Although I agree that there probably is an effect of pseudoneglect, I expect that most Western participants use a strategy starting at the left side, and working their way towards the right side, in line with the culturally defined habitual reading direction (left-to-right). I am aware that there are many studies showing that a leftward bias exists even when there is no reason to believe that reading direction would affect this (e.g., if there is no search or scanning involved), and that there are studies showing that people who are used to a right-to-left reading direction still show a leftward bias, but specifically for the cancellation task I can imagine that reading direction and strategy will have such a large influence that it might overrule the more subtle leftward bias. Especially since at some point there simply is no time left, and participants cannot complete their search towards the right side of the template, missing the targets there. Can the authors comment on this, and include some literature on effects of reading direction on visual search? In the current study, did participants typically reach the right side of the template? Was there a relation between the CoC and the starting point (left or right)? This will provide insight into the effect of the starting point on the CoC. Furthermore, the CoC values are very high (values below -0.25 are not uncommon) which further stresses that no subtle left-right difference is reflected, but more likely a much larger effect of strategy.

Minor points

- Please add the number of participants in the abstract

- Is there information on the test-retest reliability in healthy controls for the tests that are used? It would be useful to include this information.

- Report the statistical results, including exact p-values and effect sizes, for all the post-hoc tests

- Why was only the Mesulam cancellation test correlated with the LQ of the Cloud task, and not the other cancellation tests?

- The figures are informative, good that the individual data points are presented. However, it would be more complete to include figures for all the other outcome variables. In addition, it would be informative to also include the group means and error bars in the graphs.

- The figure captions can include more information, such as the number of participants per group, and what the horizontal line at “0” indicates.

Good luck with the research,

Teuni ten Brink

Reviewer #2: Geers and colleagues investigated the effect of anodal tDCS applied over left or right parietal cortex on visual search and imagery abilities in a sample of healthy participants. The authors reported no effect of tDCS on neither visual search, visual imagery nor visuospatial bias, thus concluding that anodal tDCS of the parietal cortex was not sufficient to alter the exploration of visual and mental images and to counteract spatial bias in visual search.

I have some major issues mainly regarding the methodology adopted to investigate pseudoneglect, the sample size calculation and the statistical analysis performed that prevent me to endorse the paper for publication in its present form. Below my main concerns.

- Even if cancellation tests represent a clinical gold standard for assessing neglect, the way it is scored in the present study does not allow to fully capturing the potential modulatory effect on visuospatial asymmetries. Indeed, with a 30 sec time limit, it is possible that right side omissions will always be greater to left side omissions because of the cultural left-to-right reading and writing habits. Indeed, native language direction modulates spatial bias having an effect on the side of the omissions (e.g. Rinaldi et al., 2014). Furthermore, in visual search and cancellation tasks a preference to direct the first eye movement in the upper left quadrant has been reported (Behrmann, Watt, Black, & Barton, 1997; Weintraub & Mesulam, 1988). Accordingly, Table 1 clearly indicates a greater number of crossed targets in the left visual field in all conditions. In my opinion, the methods implemented in the present study has the merit of showing the impact of tDCS on visual search (i.e., cancellation of targets among distractors) but they do not represent an adequate model to measure its effect on pseudoneglect.

-It is not clear to me how effect size for sample size calculation has been obtained. Indeed, authors provided no information on the presumed effect size. Based on which findings the authors considered a medium effect size (f=0.25)? Considering the concerns of reliability of tDCS effects, mainly due to reduced replicability and to interindividual variability in responses, sample size estimation is crucial, with underpowered studies being a potential reason for the reported inconsistences.

- Authors wrote that statistical analysis was performed on the percentage of crossed targets but Table 1 and Supporting Information 1 report the mean number of crossed targets. Please clarify this point.

- Post-hoc comparisons to investigate the significant “test” x “tDCS” x “hemisphere” interaction have been performed by comparing the number of crossed targets (or percentage?) between anodal and sham tDCS, not considering the “hemisphere” factor. Did authors performed an ANOVA with the factors “tDCS” and “hemisphere”, separately for each cancellation task?

- No mention is made regarding data distribution. Did authors check for normality of data distribution before performing parametric statistics?

-In the discussion section, authors did not consider all the tDCS-related factors possibly affecting tDCS effects on behavioral performance (e.g. intensity, montage, online vs offline protocols, state-dependency, baseline capacity, etc.). Please consider adding these considerations in the discussion.

Minor issues:

- No mention is made regarding handedness. Please provide this information and whether it has been assessed with Edinburgh Handedness questionnaire.

- How was randomization (i.e., left vs right hemisphere stimulation) performed?

- A table with CoC results is missing.

- P-values reported in the manuscript are not consistent with those reported in the Supporting Information (e.g., p. 12 line 263: p=0.025; supporting information 2: p=0.041).

- The authors applied an online tDCS protocol, with participants performing tasks within the 20 minutes of stimulation. How long after the start of the stimulation did the tasks begin? How long did the tasks last? Please clarify this point.

6. PLOS authors have the option to publish the peer review history of their article (what does this mean?). If published, this will include your full peer review and any attached files.

Reviewer #1: No

Reviewer #2: No

---

## [Author Response · Author response to Decision Letter 0]

30 Aug 2024

We thank both reviewers for their insightful comments that draw our attention on the importance of the starting point. Beyond its influence on the visual search strategy, the starting point has proved to be a relevant dependent variable we had initially overlooked. In response to the reviewers’ questions, we extracted the average x-coordinates of the first mark and then had the idea to further investigated the effect of tDCS on the position of the first cancelled target, which is known to be a good indicator of neglect in patients. The results showed that anodal stimulation of the parietal cortex resulted in a significant shift of the position of the first cancelled target by nearly 100 pixels in the contralateral direction. This finding suggest that participants started from a different position as a result of a hemispheric imbalance induced by tDCS. This finding has been incorporated in the revised manuscript and, given its scientific and clinical significance, it has also been reflected in the title. Note that we have kept track of all the reviewers’ remarks in the revised manuscript where this new finding is put into perspective with other aspects of the results.

Reviewer #1

In two sessions, either tDCS or sham stimulation was applied in healthy participants, on the left hemisphere (n = 23) or right hemisphere (n = 23). Effect of the stimulation on visual search and visual imagery was evaluated, specifically focusing on the visuospatial asymmetry in performance. Anodal tDCS was applied over the parietal cortex, increasing the spontaneous firing rate, assumed to improve visual detection in the contralateral hemifield. The study is relevant to inform treatments of neglect, and conducted properly. I do have some suggestions that the authors may want to take into account.

1. The authors mention that the cancellation tasks are sensitive to pseudoneglect, especially since a time limit was used. Although I agree that there probably is an effect of pseudoneglect, I expect that most Western participants use a strategy starting at the left side, and working their way towards the right side, in line with the culturally defined habitual reading direction (left-to-right). I am aware that there are many studies showing that a leftward bias exists even when there is no reason to believe that reading direction would affect this (e.g., if there is no search or scanning involved), and that there are studies showing that people who are used to a right-to-left reading direction still show a leftward bias, but specifically for the cancellation task I can imagine that reading direction and strategy will have such a large influence that it might overrule the more subtle leftward bias. Especially since at some point there simply is no time left, and participants cannot complete their search towards the right side of the template, missing the targets there.

1a. Can the authors comment on this, and include some literature on effects of reading direction on visual search?

The reviewer is right to highlight the potential influence of reading habits on our findings. Several studies have shown that most adults cancel the first target on the left side (1–4). This leftward bias was not present in young children (around 3 years old) but increased with age, suggesting that it is learned through reading habits (5). Consistent with this idea, the starting point for target cancellation is influenced by reading direction. Left-to-right readers (e.g., Italians) were found to start more often on the left side, whereas right-to-left readers (e.g., Israelis) showed no bias or a slight rightward bias. However, the initial rightward bias of right-to-left readers did not simply mirror the leftward bias of left-to-right readers, suggesting a moderating effect of neurobiological asymmetries that orient visuospatial attention to the left side of space according to Rinaldi and colleagues (6).

In our study, we also observed an initial leftward bias, with the average (± SD) x-coordinate of the first cancellation mark being 121 (± 323) pixels on the left of the center of the template. We cannot exclude that part of this bias arose from the reading habits of our participants, who were left-to-right readers. However, we found that this left initial bias was modulated by tDCS, so that it was enhanced (more lateral) during anodal stimulation of the right parietal cortex and reduced (more central) during anodal stimulation of the left parietal cortex. Notably, tDCS shifted the starting position by nearly 100 pixels, compared to sham condition, suggesting that participants selected a more lateral or central target as a result of the hemispheric imbalance caused by tDCS. The two-way interaction between CONDITION and HEMISPHERE, χ²(1) = 6.75, p = .009, was integrated in a three-way interaction with TEST χ²(2) = 8.88, p = .012, indicating that this effect was observed in the Mesulam test and, to a lesser extent, in the Star test, but not in the Ota test (see detailed results on pages 13-15). Hence, our results suggest that at least part of the left initial bias is due to neurobiological asymmetries that can be modulated with tDCS. 

We have added literature about the influence of reading habits on the initial bias in visual search in the Introduction section (page 5), the described analysis in the Methods (pages 10-11) and Results (pages 13-15) sections and discussed them together in the Discussion section (page 21). 

1b. In the current study, did participants typically reach the right side of the template?

Yes, participants crossed the midline in most trials, regardless of whether they started from the left or right side of the template. This occurred, on average, 11.7 ± 6.8 seconds after the trial onset, i.e., before the middle of the trial whose total duration was 30 seconds, indicating that the lower proportion of targets crossed on the right side cannot be attributed to a lack of time. This pattern is also visible when computing a center of cancellation (CoC) for each 5-second time bin. As shown in the figure below, regardless of whether the first target cancelled was on the right or left, the CoC progressively shifted towards the opposite direction and changed sign before the middle of the trial (between bins 3 and 4). This strategy left enough time to cross as many targets as on the other side during the remaining time. However, it is noticeable that the CoC is less polarized during the second half of the trial (absolute value < 0.4) than the first half (absolute value > 0.5), indicating that participants did not go as far on the side opposite to the starting side. The most reasonable explanation is that participants did not progress linearly towards the other side but still crossed some targets on the starting side due to loops and revisits of already explored locations, as visible in the individual data that have been added in the Open Science Framework (https://osf.io/4dbs5/). This is now discussed on page 21 of the manuscript. 

1c. Was there a relation between the CoC and the starting point (left or right)? This will provide insight into the effect of the starting point on the CoC.

Yes, we found a significant positive relationship between the CoC and x-coordinate of the first cancellation mark in the sham condition. Specifically, a more leftward starting position resulted in a more leftward CoC, χ²(1) = 27.97, p < .001. This relationship persisted in the active tDCS condition, χ²(1) = 53.65, p < .001, despite the fact that tDCS modulated the initial leftward bias and not the CoC. Thus, the processes that sustain the visual search are partially distinct from those that guide the selection of the first cancelled target as it is possible to modulate the starting position independently of the CoC. Importantly, as mentioned previously, this result is not due to a lack of time to visit both sides of the template. This is now discussed on page 21 of the manuscript. 

1d. Furthermore, the CoC values are very high (values below -0.25 are not uncommon) which further stresses that no subtle left-right difference is reflected, but more likely a much larger effect of strategy.

High CoC values have indeed been observed in some individuals and we acknowledge that the CoC might have been enhanced by the strategy of the participants. An intricate question now discussed in the paper is to which extent the adopted strategy reflects neurobiological (i.e. a left-right hemisphere imbalance) or cultural (i.e. reading habits) factors (page 21). As pseudoneglect is generally associated with a left-right hemisphere imbalance (7), we decided to replace the term “pseudoneglect” by “leftward bias” to not bias the conclusions to one or the other account.

Minor points

- Please add the number of participants in the abstract. 

The number of participants has been added in the abstract.

- Is there information on the test-retest reliability in healthy controls for the tests that are used? It would be useful to include this information.

Cancellation tests typically exhibit high test-retest reliability regarding the number of targets crossed (8,9). In the current study, the experimental design was not ideally suited for assessing test-retest reliability due to the use of different (i.e., flipped) versions of the tests across repetitions. Despite this limitation, we calculated Pearson correlation coefficients to evaluate the consistency of each dependent variable between the normal (first) and flipped (second) versions of each test during the sham session. For the x-coordinate of the first cancelled target, we observed a significant positive correlation for each test (Mesulam: r = 0.38, p = .007; Ota: r = .72, p < .001; Star: r = .72, p < .001). Regarding the CoC, a significant correlation was found only for the Star test, r = .37, p = .011, whereas no significant correlations were observed for the Mesulam, r = .01, p = .948, and Ota, r = -.18, p = 0.211, tests. Finally, for the percentage of crossed targets, positive correlations were evident for the Mesulam, r = 0.63, p <.001, Ota, r = 0.70, p < .001, and Star, r = 0.60, p < .001, tests. These results align with previous findings, supporting the good test-retest reliability of cancellation tests, except for the CoC in the Mesulam and Ota tests. These results have been added to the manuscript (page 18) but should however be interpret with caution, as the reliability was assessed using flipped versions of the tests.

- Report the statistical results, including exact p-values and effect sizes, for all the post-hoc tests.

This has been addressed in the revised manuscript. 

- Why was only the Mesulam cancellation test correlated with the LQ of the Cloud task, and not the other cancellation tests?

The correlation is less reliable for the Star and Ota cancellation tasks due to a ceiling effect in the number of crossed targets, reducing the sensitivity of these tasks (i.e. less inter-individual variability). However, for the sake of completeness, we now report the correlation for these two tasks as well, which were also not significant (page 19).

- The figures are informative, good that the individual data points are presented. However, it would be more complete to include figures for all the other outcome variables. In addition, it would be informative to also include the group means and error bars in the graphs.

We now provide figures for all the outcome variables, including initial starting position, with the mean group performance and related standard error superimposed (in black) on the individual data (in light grey).

- The figure captions can include more information, such as the number of participants per group, and what the horizontal line at “0” indicates.

This information has been added in the revised manuscript. Thank you for drawing our attention on the missing information.

Reviewer #2

Geers and colleagues investigated the effect of anodal tDCS applied over left or right parietal cortex on visual search and imagery abilities in a sample of healthy participants. The authors reported no effect of tDCS on neither visual search, visual imagery nor visuospatial bias, thus concluding that anodal tDCS of the parietal cortex was not sufficient to alter the exploration of visual and mental images and to counteract spatial bias in visual search.

I have some major issues mainly regarding the methodology adopted to investigate pseudoneglect, the sample size calculation and the statistical analysis performed that prevent me to endorse the paper for publication in its present form. Below my main concerns.

1. Even if cancellation tests represent a clinical gold standard for assessing neglect, the way it is scored in the present study does not allow to fully capturing the potential modulatory effect on visuospatial asymmetries. Indeed, with a 30 sec time limit, it is possible that right side omissions will always be greater to left side omissions because of the cultural left-to-right reading and writing habits. Indeed, native language direction modulates spatial bias having an effect on the side of the omissions (e.g. Rinaldi et al., 2014). Furthermore, in visual search and cancellation tasks a preference to direct the first eye movement in the upper left quadrant has been reported (Behrmann, Watt, Black, & Barton, 1997; Weintraub & Mesulam, 1988). Accordingly, Table 1 clearly indicates a greater number of crossed targets in the left visual field in all conditions. In my opinion, the methods implemented in the present study has the merit of showing the impact of tDCS on visual search (i.e., cancellation of targets among distractors) but they do not represent an adequate model to measure its effect on pseudoneglect.

We would like to thank the reviewer for drawing our attention on the importance of the starting point, which allowed us to maximize the insights from our data. Several studies have indeed reported that most adults cancel the first target on the left side of the template (1–4). This leftward bias was absent in young children (around 3 years old) but increased with age, suggesting that it develops through reading habits (5). Consistent with this idea, the starting point for target cancellation is influenced by reading direction. Left-to-right readers (e.g., Italians) were found to start more often on the left side, whereas right-to-left readers (e.g., Israelis) showed no bias or a slight rightward bias. However, the initial rightward bias of right-to-left readers did not simply mirror the leftward bias of left-to-right readers, suggesting a moderating effect of neurobiological asymmetries that orient visuospatial attention to the left side of space according to Rinaldi and colleagues (6). 

To determine whether such a bias in the starting position was present in our study, we extracted the average x-coordinates of the first marks, which were, on average, 121 pixels to the left of the center of the template. We then had the idea to further investigated the effect of tDCS on the position of the first cancelled target, which is known to be a good indicator of neglect in patients. The results showed that anodal stimulation of the parietal cortex caused a significant shift of nearly 100 pixels in the contralateral direction for the first cancelled target. This finding suggest that participants started from a different position due to the interhemispheric imbalance induced by tDCS. The two-way interaction between CONDITION and HEMISPHERE, χ²(1) = 6.75, p = .009, was integrated in a three-way interaction with TEST χ²(2) = 8.88, p = .012, indicating that this effect was observed in the Mesulam test and, to a lesser extent, in the Star test, but not in the Ota test (see detailed results on pages 13-15). We cannot exclude the possibility that part of this bias arises from our participants being left-to-right readers, as it is challenging to disentangle the effect of reading habits from neurobiological asymmetries. However, the observed effect of tDCS suggests that this initial leftward bias cannot be entirely attributed to reading habits and that interhemispheric balance, which can be modulated through parietal stimulation, also plays a role.

Furthermore, the leftward bias in the spatial distribution of cancelled targets (or C

---

## [Decision Letter · Decision Letter 1]

14 Nov 2024

PONE-D-24-02796R1Modulation of initial leftward bias in visual search by parietal tDCSPLOS ONE

Dear Dr. Geers,

Thank you for submitting your manuscript to PLOS ONE. After careful consideration, we feel that it has merit but does not fully meet PLOS ONE’s publication criteria as it currently stands. Therefore, we invite you to submit a revised version of the manuscript that addresses the points raised during the review process.

We look forward to receiving your revised manuscript.

Kind regards,

Abdolvahed Narmashiri

Academic Editor

PLOS ONE

Journal Requirements:

Reviewers' comments:

Reviewer's Responses to Questions

**Comments to the Author**

1. If the authors have adequately addressed your comments raised in a previous round of review and you feel that this manuscript is now acceptable for publication, you may indicate that here to bypass the “Comments to the Author” section, enter your conflict of interest statement in the “Confidential to Editor” section, and submit your "Accept" recommendation.

Reviewer #1: (No Response)

Reviewer #2: All comments have been addressed

2. Is the manuscript technically sound, and do the data support the conclusions?

Reviewer #1: Partly

Reviewer #2: Yes

3. Has the statistical analysis been performed appropriately and rigorously? 

Reviewer #1: Yes

Reviewer #2: Yes

4. Have the authors made all data underlying the findings in their manuscript fully available?

Reviewer #1: Yes

Reviewer #2: Yes

5. Is the manuscript presented in an intelligible fashion and written in standard English?

Reviewer #1: Yes

Reviewer #2: Yes

6. Review Comments to the Author

Reviewer #1: I appreciate the changes that the authors have made. The description of what the bias entails is now more accurate, and analyzing the first target position seems appropriate. I have some suggestions that the authors may want to consider.

It would be transparent to mention in the end of the introduction (and maybe also the methods and results) that the CoC was the primary endpoint, and later on secondary analyses were conducted on the starting position and total percentage of cancelled targets. Also reflect on this in the discussion section. The evidence is still valuable, but since the analysis differed from the planned analyses, the results (especially the marginal effect on the total percentage of cancelled targets in just one task) should be interpreted with caution.

Related to the previous point, the authors stress the ‘marginally significant’ finding in the effect on the percentage of targets, whereas the non-significant findings are discussed less. I think the authors should be more cautious in the interpretation of this finding.

The authors use the x-position in pixels, which is fine for the analysis, but it would be insightful to add in the results section how much the x-position shifted in terms of visual degrees or cm. Also provide the size of the laptop screen in cm.

Reviewer #2: The authors have addressed all the reviewer's comments. I appreciate the authors’ effort in addressing the concerns previously raised, which have significantly improved the manuscript quality making it suitable for publication. I still have few minor concerns to be addressed. Please comment on the lack of a significant visual imagery effect, which may be attributable to the fact that the Cloud task is always administered after the visual search tasks. Furthermore, no mention is made regarding the visual imagery findings in the abstract. Please consider to integrate this.

7. PLOS authors have the option to publish the peer review history of their article (what does this mean?). If published, this will include your full peer review and any attached files.

Reviewer #1: **Yes: **Teuni ten Brink

Reviewer #2: No

---

## [Author Response · Author response to Decision Letter 1]

20 Nov 2024

> We thank both reviewers for their positive feedback and for recognizing the improvements we have made. We genuinely appreciated their thoughtful suggestions and carefully considered them to further enhance the manuscript. Their input has been invaluable in refining our work.

Reviewer #1: I appreciate the changes that the authors have made. The description of what the bias entails is now more accurate, and analyzing the first target position seems appropriate. I have some suggestions that the authors may want to consider.

It would be transparent to mention in the end of the introduction (and maybe also the methods and results) that the CoC was the primary endpoint, and later on secondary analyses were conducted on the starting position and total percentage of cancelled targets. Also reflect on this in the discussion section. The evidence is still valuable, but since the analysis differed from the planned analyses, the results (especially the marginal effect on the total percentage of cancelled targets in just one task) should be interpreted with caution.

> We appreciate the importance of transparency and have clarified in both the Introduction (page 6, lines 123-128), Methods (page 11, lines 253-255) and Discussion (page 23, lines 501-503) that analyses on the starting position were secondary analyses. However, we would like to emphasize that analyses on the percentage of cancelled targets were planned from the start and were included in the first version of the manuscript. In the initial version, we observed a significant tDCS × hemisphere × task interaction on overall performance. Following a reviewer’s suggestion to address normality concerns, we applied a mixed-model analysis instead of an ANOVA, which resulted in a slightly different outcome—a significant tDCS × hemisphere interaction, with a marginal difference between active and sham tDCS in the right hemisphere group. Importantly, we have carefully discussed these results, particularly considering the marginal significance of the post-hoc test, to ensure an appropriately cautious interpretation (pages 22-23).

Related to the previous point, the authors stress the ‘marginally significant’ finding in the effect on the percentage of targets, whereas the non-significant findings are discussed less. I think the authors should be more cautious in the interpretation of this finding.

> We agree and have revised the manuscript to adopt a more cautious tone when interpreting the marginally significant finding on the percentage of cancelled targets (page 22-23). 

The authors use the x-position in pixels, which is fine for the analysis, but it would be insightful to add in the results section how much the x-position shifted in terms of visual degrees or cm. Also provide the size of the laptop screen in cm.

> Thank you for pointing this out. We have added the information on the shift in x-position in terms of centimeters in the Results section (pages 13-14). Additionally, we have included the size of the tablet screen in in the Methods section to provide the necessary context for interpreting these results (page 7, lines 173-174).

Reviewer #2: The authors have addressed all the reviewer's comments. I appreciate the authors’ effort in addressing the concerns previously raised, which have significantly improved the manuscript quality making it suitable for publication. I still have few minor concerns to be addressed. Please comment on the lack of a significant visual imagery effect, which may be attributable to the fact that the Cloud task is always administered after the visual search tasks. Furthermore, no mention is made regarding the visual imagery findings in the abstract. Please consider to integrate this.

> We have added a discussion on the lack of a significant of the visual imagery effect and its potential relation to task order in the Discussion section (page 24, lines 531-541). Additionally, we have integrated the visual imagery findings into the Abstract (page 2, lines 40-44).

---

## [Decision Letter · Decision Letter 2]

1 Dec 2024

Modulation of initial leftward bias in visual search by parietal tDCS

PONE-D-24-02796R2

Dear Dr. Geers,

We’re pleased to inform you that your manuscript has been judged scientifically suitable for publication and will be formally accepted for publication once it meets all outstanding technical requirements.

Kind regards,

Abdolvahed Narmashiri

Academic Editor

PLOS ONE

Additional Editor Comments (optional):

Reviewers' comments:

Reviewer's Responses to Questions

**Comments to the Author**

1. If the authors have adequately addressed your comments raised in a previous round of review and you feel that this manuscript is now acceptable for publication, you may indicate that here to bypass the “Comments to the Author” section, enter your conflict of interest statement in the “Confidential to Editor” section, and submit your "Accept" recommendation.

Reviewer #1: All comments have been addressed

2. Is the manuscript technically sound, and do the data support the conclusions?

Reviewer #1: Yes

3. Has the statistical analysis been performed appropriately and rigorously? 

Reviewer #1: Yes

4. Have the authors made all data underlying the findings in their manuscript fully available?

Reviewer #1: Yes

5. Is the manuscript presented in an intelligible fashion and written in standard English?

Reviewer #1: Yes

6. Review Comments to the Author

Reviewer #1: (No Response)

7. PLOS authors have the option to publish the peer review history of their article (what does this mean?). If published, this will include your full peer review and any attached files.

Reviewer #1: No

---

## [Editor Report · Acceptance letter]

11 Dec 2024

PONE-D-24-02796R2 

PLOS ONE

Dear Dr. Geers, 

I'm pleased to inform you that your manuscript has been deemed suitable for publication in PLOS ONE. Congratulations! Your manuscript is now being handed over to our production team.

Kind regards, 

on behalf of

Dr. Abdolvahed Narmashiri 

Academic Editor

PLOS ONE